# Evaluating the antibody response to SARS-COV-2 vaccination amongst kidney transplant recipients at a single nephrology centre

Chukwuma A. Chukwu[1,2]*, Kassir Mahmood[1], Safa Elmakki[1], Julie Gorton[1], Phillip A. Kalra[1,2], Dimitrios Poulikakos[1,2], Rachel Middleton[1]

1 Department of Nephrology, Salford Royal Hospital, Northern Care Alliance NHS Foundation Trust, Salford, United Kingdom, 2 Faculty of Biology, Medicine and Health, University of Manchester, Manchester, United Kingdom

* chukwuma.chukwu@nca.nhs.uk

**Data Availability Statement:** All relevant data are within the manuscript and its Supporting Information files.

## Abstract

### Background and objectives

Kidney transplant recipients are highly vulnerable to the serious complications of severe acute respiratory syndrome coronavirus 2 (SARS-COV-2) infections and thus stand to benefit from vaccination. Therefore, it is necessary to establish the effectiveness of available vaccines as this group of patients was not represented in the randomized trials.

### Design, setting, participants, & measurements

A total of 707 consecutive adult kidney transplant recipients in a single center in the United Kingdom were evaluated. 373 were confirmed to have received two doses of either the BNT162b2 (Pfizer-BioNTech) or AZD1222 (Oxford-AstraZeneca) and subsequently had SARS-COV-2 antibody testing were included in the final analysis. Participants were excluded from the analysis if they had a previous history of SARS-COV-2 infection or were seropositive for SARS-COV-2 antibody pre-vaccination. Multivariate and propensity score analyses were performed to identify the predictors of antibody response to SARS-COV-2 vaccines. The primary outcome was seroconversion rates following two vaccine doses.

### Results

Antibody responders were 56.8% (212/373) and non-responders 43.2% (161/373). Antibody response was associated with greater estimated glomerular filtration (eGFR) rate [odds ratio (OR), for every 10 ml/min/1.73m$^2$ = 1.40 (1.19–1.66), P<0.001] whereas, non-response was associated with mycophenolic acid immunosuppression [OR, 0.02(0.01–0.11), p<0.001] and increasing age [OR per 10year increase, 0.61(0.48–0.78), p<0.001]. In the propensity-score analysis of four treatment variables (vaccine type, mycophenolic acid, corticosteroid, and triple immunosuppression), only mycophenolic acid was significantly associated with vaccine response [adjusted OR by PSA 0.17 (0.07–0.41): p<0.001]. 22 SARS-COV-2 infections were recorded in our cohort following vaccination. 17(77%)

**Funding:** The author(s) received no specific funding for this work.

**Competing interests:** The authors have declared that no competing interests exist.

infections, with 3 deaths, occurred in the non-responder group. No death occurred in the responder group.

## Conclusion

Vaccine response in allograft recipients after two doses of SARS-COV-2 vaccine is poor compared to the general population. Maintenance with mycophenolic acid appears to have the strongest negative impact on vaccine response.

## Introduction

The effects of coronavirus disease 2019 (COVID -19) have resulted in more than 190 million infections and more than 4 million deaths worldwide [1].

Kidney transplant recipients (KTR) are among the most vulnerable to the complications of COVID-19 infections [2] and thus stand to benefit the most from any preventive intervention such as vaccination. However, while COVID-19 vaccine trials have shown excellent efficacy in the general population, KTR have largely been excluded from these studies meaning that the protective effects of vaccination have not been thoroughly investigated in these patients [3]. Regrettably, recent real-world evidence suggests a sub-optimal antibody response by KTR to the currently deployed severe acute respiratory syndrome coronavirus 2 (SARS-CoV-2) vaccines. The reported seroconversion rates range from 0–17% after one vaccine dose and 3–59% after two doses of the mRNA vaccines [3]. Furthermore, the estimated pooled seroconversion rates among KTR are 8% after one vaccine dose and 35% after the two doses [3].

There have also been multiple reports of the occurrence of COVID-19 disease after complete vaccination, in some cases sadly resulting in death [4, 5]. Recent studies appear to suggest that these cases of severe COVID-19 infections after complete vaccination have occurred in individuals with low or absent antibody response to the vaccine [5–7].

Few studies have explored the factors associated with inadequate antibody response in KTR. Understanding the antibody response rates and the factors that influence antibody response in KTR will improve risk stratification and inform vaccination development and deployment in this vulnerable group.

This study sought to investigate the antibody response rate to 2 doses of SARS-COV-2 vaccine in a single center cohort of KTR and identify factors associated with inadequate antibody response. We also followed up the KTR population for COVID-19 infections following vaccination.

## Materials and methods

We carried out a retrospective observational cohort study of prevalent COVID naïve kidney transplant recipients at our tertiary nephrology center, who were vaccinated with either of the two main UK approved COVID-19 vaccines (BNT162b2/Pfizer-BioNTech or AZD1222/ChAdOx1 nCoV-19/Oxford-Astra-Zeneca vaccines).

### Study population

The study population consisted of all adult kidney transplant recipients (n = 707) with a functioning transplant (defined as those not receiving maintenance dialysis therapy post transplantation) who were under follow up at our nephrology center.

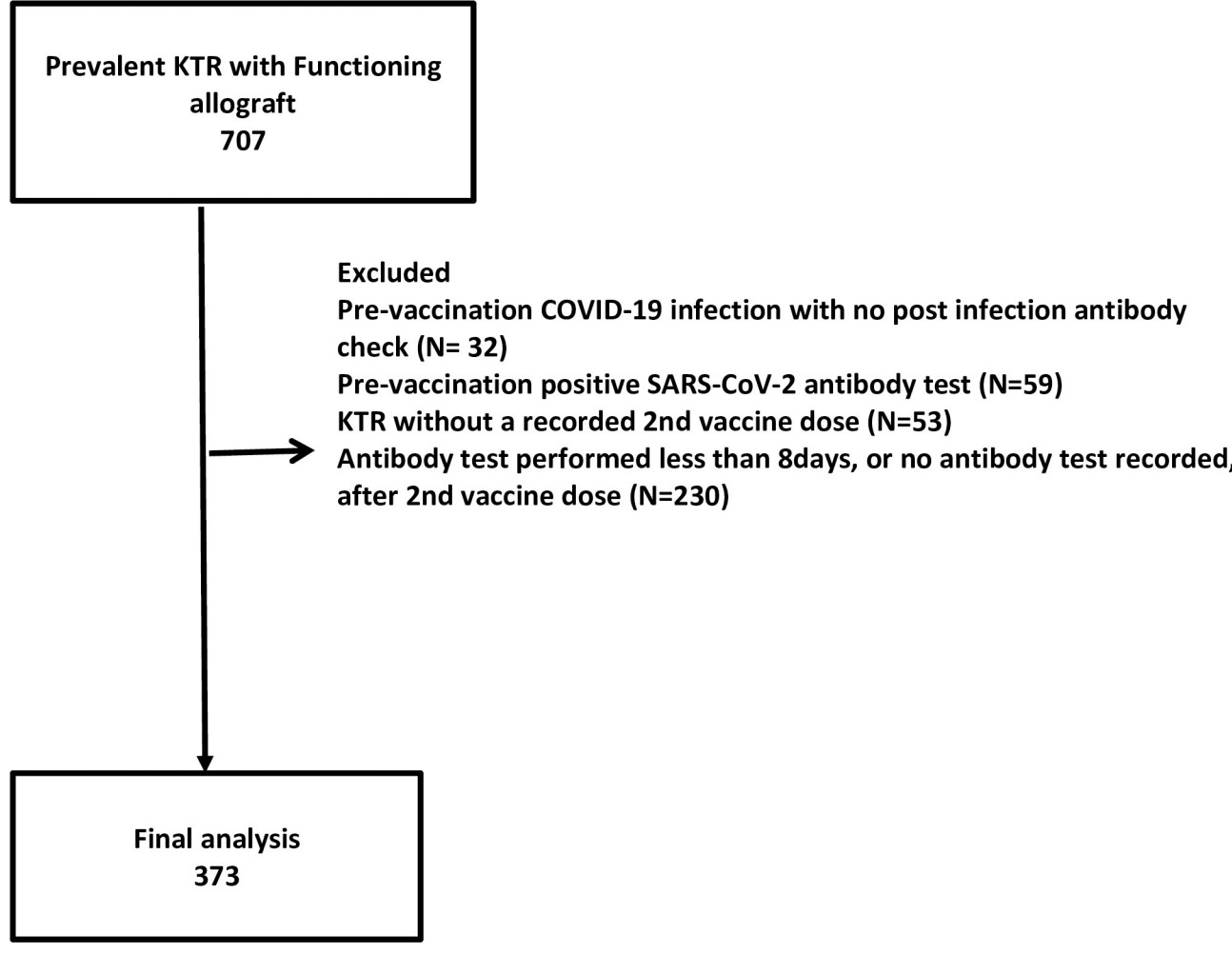

**Fig 1. Cohort selection flow chart.** COVID-19, coronavirus disease-2019; KTR, kidney transplant recipients; SARS-COV-2, severe acute respiratory syndrome coronavirus 2.

**Study subjects (see Fig 1).** In the final analysis, we included KTR have had two doses of the above-named vaccines between December 2020 and July 2021. Also, they would have had a post-vaccination antibody assay at a minimum of eight days post-vaccination. KTR who had a confirmed history of SARS-CoV-2 infection before vaccination were excluded from analysis, as were those with a positive SARS-COV-2 antibody test pre-vaccination. Fig 1 shows the flow chart for participant selection.

At the time of data collection, two SARS-CoV-2 vaccines had predominantly been used for the UK's national vaccination programme. They were the Pfizer-BioNTech (BNT162b2) vaccine and the Oxford-AstraZeneca ChAdOx1-S (AZD1222) vaccine. A third vaccine, the mRNA-1273 developed by Moderna inc had been authorized for use in the UK but had not been made widely available; only a few of the KTR had received the first dose of the Moderna vaccine and so were not included in the final analysis.

In line with the UK Joint Committee on Vaccination and Immunization (JCVI) guidelines, the vaccine dosing interval between December 2020 and February 2021 was 21–28 days (about 4 weeks). This was increased between March and May 2021 to 77–84 days (about 3 months).

Subsequently, in late May 2021, as the delta variant became prevalent, the dosing interval was reduced to 56–63 days (about 2 months) to optimize coverage.

## Ethical considerations

The study was carried out according to the Declaration of Helsinki. Approval for the study was obtained from the Research and Innovation department of the Northern care Alliance NHS group (Ref: S21HIP51). As this was a retrospective observational study with complete patient anonymity, it was deemed unnecessary to obtain written consent from patients.

## Data collection

Data were collected from the electronic patient records. These included demographic data, comorbidities including primary renal disease, cardiovascular diseases (CVD) and diabetes. CVD was a composite of heart failure, coronary artery disease, myocardial infarction, non-fatal cardiac arrest, stroke, or peripheral vascular disease. Dates of vaccination and the type of SARS-CoV-2 vaccine were recorded. If the date of the second vaccination was not recorded in the subject's hospital records because the vaccine was delivered in the community, the dosing interval was calculated based on JCVI recommended vaccination interval. Only thirteen subjects had their dosing interval estimated this way. The type and number of immunosuppressive medications, serum creatinine measured around the latest vaccine dose and the estimated glomerular filtration rate (eGFR) were also collected. Post vaccine Anti SARS-COV-2 antibody assay was performed concurrently with routine blood tests, including eGFR. eGFR calculation was based on the Modification of Diet in Renal Disease (MDRD) equation. The primary outcome was the rate of seropositivity for the anti-SARS-COV-2 antibody. Information regarding COVID-19 infection and COVID-19 associated deaths post vaccination was also collected until 31st August 2021.

## Serological testing

Anti-SARS-COV-2 antibody testing was performed at routine clinic appointments as per standard clinical practice in our department. This was carried out using the Public Health England approved Siemens Atellica-IM SARS-CoV-2 immunoassay, which is targeted to identify antibodies against viral spike protein (anti-S) receptor binding domain (S1 RBD) [8–10].

The Atellica SARS-CoV-2 Total (COV2T) assay is a sandwich immunoassay which uses the acridinium ester chemiluminescent technology to detect antibodies to SARS-CoV-2 in human serum and plasma. It contains a preformed complex of streptavidin-coated microparticles and biotinylated SARS-CoV-2 recombinant antigens, which captures anti-SARS-CoV-2 antibodies in a patient's sample. The antibodies are then highlighted by a light reagent containing acridinium-ester-labelled SARS-CoV-2 recombinant antigens.

Antibody tests were reported as index values and considered positive (reactive) if the index value was >1.0 and negative (non-reactive) if <1.0. Atellica SARS-CoV-2 assay has a good sensitivity and specificity (98.1 and 99.9 respectively) [8–10].

## Statistical analysis

Data are summarized as means ± standard deviation, medians (interquartile range) or number (%). A comparative univariate analysis (UVA) was conducted between the vaccine response and the non-response group using Chi-square test and t-test or non-parametric Wilcoxon rank-sum test to assess statistical significance. Multivariate regression analyses (MVA) and a

propensity score-matched analysis (PSA) were performed to explore the factors associated with antibody response.

In the MVA, Variables were chosen by purposeful selection [11]. The variables with a p-value ≤ 0.25 in the UVA were used to build the initial MVA model. Variables were then removed from the model in a stepwise manner if they were either non-significant at alpha level of 0.1 or non-confounders (i.e., exclusion of the variable did not result in a >20% change in the parameter estimates of remaining variables). This process was repeated until only variables with alpha ≤0.1 or confounders were left in the model. In the next stage, variables from the UVA that were not part of the initial MVA model were added one at a time into the MVA and retained in model if they were either significant at an alpha level of 0.1 or resulted in a >20% change in the parameter estimates of other covariates in the model.

To further control for a broader range of confounding variables, a PSA was also conducted to create comparable risk groups between participants to assess the effect of MPA, steroids, triple immunosuppression, and vaccine type on vaccine response. Participants were matched on baseline demographic and clinical characteristics including age, sex, BMI, ethnicity (categorized into black and ethnic minority ethnic groups (BAME) and non-BAME), primary renal disease (6 categories), history of CVD, diabetes, donor type (life or cadaveric donor), number of HLA, A, B and DR mismatches, antimetabolite immunosuppression, CNI immunosuppression, steroid and eGFR.

We estimated propensity scores (PS) using the listed covariates for each of the four treatment variables. A separate analysis was carried out for each treatment variable.

Treated and untreated participants were matched on the PS within a calliper's width of 0.2 standard deviations of the logit PS.

We used a method based on the standardized mean difference of the observed covariates between both groups to ensure a good match between treated and untreated subjects. The standardized differences for both the raw and matched samples were calculated and compared between the treated and untreated groups with a target mean standardized difference of <10% after matching. The PSA enables us to widen the pool of confounding factors to include variables like the number of HLA mismatches and allograft types. Covariate balance was achieved in the PSA as evidenced by a mean standardized difference of less than 10% for each evaluated treatment variable. The number of subjects included in each PS generation and the matched samples respectively and standardized differences as shown in Table 1. Adjusted odds ratios and confidence intervals were then calculated by regressing vaccine response rates on each treatment variable in the matched groups. Statistical analysis was carried out using Stata statistical software version 14 (Statacorp LP. College Station, Tx, USA) licensed to the University of Manchester.

**Table 1. Summary of propensity matching.**

| Treatment variables | Mean standardized differences (%) | | Observations within common support (N) | | Treated (N) | | Control (N) | |
|---|---|---|---|---|---|---|---|---|
| | Raw | Matched | Raw | Matched | Raw | Matched | Raw | Matched |
| Triple IS | 15.91 | 7.38 | 237 | 118 | 59 | 59 | 178 | 59 |
| Vaccine type[γ] | 15.20 | 9.33 | 287 | 148 | 78 | 74 | 209 | 74 |
| MPA | 31.21 | 5.54 | 237 | 224 | 188 | 188 | 49 | 36 |
| Steroids | 23.19 | 2.12 | 236 | 213 | 81 | 81 | 155 | 132 |

Standardized differences were calculated for each covariate used for generating the propensity score before and after PS matching then the average was calculated for each treatment variable target<10%; γ, the AZD1222 vaccine(treated) was compared to BNT162b2 vaccine (control); IS, immunosuppression; MPA, mycophenolic acid

## Results

Of the 707 prevalent KTR with functioning allografts, 334 were excluded (reasons shown in the flow chart), and 373 (53.3%) met the inclusion criteria for final analysis. The flow chart illustrating patient selection for the final analysis is shown in Fig 1.

72% of evaluated patients had received BNT162b2 (Pfizer-BioNTech) and 28% AZD1222 (Oxford-AstraZeneca). Antibody assays were performed at a median time of 38(22–55) days after the second dose of the vaccine. The median interval between the first and second dose of the vaccine was 77(71–84) days. The median time between kidney transplantation and vaccination was 91 (48–156) months. Of the 373 KTR included in the final analysis, 212(56.8%) were positive for SARS-COV-2 antibody (the seropositive group), and 161(43.2%) were negative (the seronegative group). The characteristics of the patients are shown in Table 2. The mean age was 55±14 years, with the seropositive group being significantly younger than the seronegative group (51±14 vs 58±13 years; P<0.001). Both groups, however, had a similar distribution of gender, ethnicity, average BMI, diabetes. There was a significantly lower prevalence of cardiovascular disease in the seropositive group (20.3% vs 30.4%: P = 0.024). The median time since transplantation was longer in the seropositive group [110(59–156) vs 72 (30–122) months: P>0.001].

Factors associated with antibody response in UVA, MVA and PSA are shown in Table 3 and Figs 2 and 3.

In the UVA, factors associated with negative antibody response included increasing age [OR 0.66/10yr increase (0.60–0.82); p<0.001, history of CVD [OR 0.58(0.36–0.93); p = 0.02] CNI [0.35 (0.13–0.95); p = 0.04, MPA [OR 0.10 (0.06–0.20); p<0.001] number of immunosuppressive agents [OR 0.58(0.39–0.86); p = 0.007], whereas the factors in favor of a positive antibody response included higher eGFR [OR per 10ml/min increase 1.12(1.01–1.25); p = 0.032], history of BK viremia [OR 2.50(1.17–5.33); p = 0.018] and history of acute rejection [OR 2.37 (1.28–4.38); p = 0.006].

In the MVA, increasing age, [OR per 10year increase 0.61, (0.48–0.78): p < 0.001] and MPA immunosuppression [OR 0.02 (0.01–0.11); P<0.001] where significantly associated with a negative vaccine response whereas higher eGFR [OR per $10mL/min/1.73 m^2$, 1.40(1.19–1.66) P<0.001] was a predictor of a positive vaccine response. We found no significant association between antibody response and gender, ethnicity, BMI, diabetes, donor type and vaccine type.

The result of the PSA confirmed the negative impact of MPA on antibody response to vaccination with an adjusted OR of 0.17(0.07–0.21); p<0,001. On the other hand, corticosteroid maintenance therapy and triple immunosuppression did not affect antibody response to vaccination.

In our cohort neither of the 2 vaccines (AZD1222 or BNT162b2) had a superior antibody response to the other [aOR 0.93 (0.41–2.13): P = 0.87].

Thirty-three SARS-COV-2 infections were recorded in our KTR population as of 31st August 2021. Among the doubly vaccinated KTR (who had a post-vaccination antibody assay) included in this study, there were 22 COVID-19 infections and 3 deaths after vaccination, of which 17(77%) infections and 3 deaths occurred in the seronegative group, whereas only 5 (23%) infections with no deaths occurred in the seropositive group.

## Discussion

In this study, in addition to an MVA, we used a PSA in a separate analysis to reduce bias from multiple confounders in this non-randomized study. We believe this provided additional validity to the result of the study.

**Table 2. Demographics, comorbidity and transplant characteristics of transplant recipients stratified by vaccine response.**

| Table 2: Characteristics according to seroconversion status | | | |
|---|---|---|---|
| | Total Cohort (n = 373) | Seropositive Response(n = 212) | Seronegative Response(n = 161) |
| Antibody status post vaccine | 373 | 212(56.8) | 161(43.2) |
| Age (Years) | 55±14 | 51±14 | 58±13 |
| Male | 228(60) | 136(64) | 92(57) |
| Caucasian | 312(83.6) | 174(82.1) | 138(85.7) |
| BAME | 60(16.1) | 38(17.9) | 22(13.7) |
| Months from transplantation | 91(48–156) | 110(59–156) | 71.5(30–122) |
| Live donor | 135(37) | 77(36.8) | 58(37.2) |
| Comorbidities | | | |
| BMI | 27.3(5.8) | 27.7± 6.1 | 26.9 ± 5.3 |
| Diabetes | 123(33) | 66(31.1) | 57(35.4) |
| CVD | 92(24.7) | 43(20.3) | 49(30.4) |
| Post-transplant cancer[α] | 24(6.4) | 12(5.7) | 12(7.5) |
| Immunosuppression | | | |
| CNI | 350(93.8) | 194(91.5) | 156(96.9) |
| MPA | 263(70.5) | 115(54.3) | 148(91.9) |
| Azathioprine | 46(12.3) | 41(19.3) | 5(3.1) |
| Corticosteroid | 134(36) | 85(40.1) | 49(30.4) |
| Triple IS | 83(22.3) | 45(21.2) | 38(23.6) |
| Dual IS | 261(70) | 139(65.6) | 122(75.8) |
| Single IS | 29(7.8) | 28(13.2) | 1(0.62) |
| eGFR [(MDRD) (ml/min/1.73m$^2$] | 47(34–60) | 49(36–63) | 43(33–56) |
| History Viral infection | | | |
| BK viremia | 38(11.2) | 28(15) | 10(6.28) |
| CMV infection | 41(12.0) | 19(10.2) | 22(14.3) |
| EBV infection | 26(7.7) | 12(6.4) | 14(9.2) |
| JCV | 9(2.7) | 4(2.1) | 5(3.3) |
| Prior Acute rejection | 60(16.1) | 44(20.8) | 16(9.9) |
| Vaccine type | | | |
| AZD1222 | 84(28) | 50(30.5) | 34(25.0) |
| BNT162b2 | 216(72) | 114(69.5) | 102(75) |
| Days from vaccine to antibody test | 38(22–55) | 38(19–54) | 39(24–55) |
| Post Vaccination COVID-19 infection | 10(2.6) | 2(1) | 8(5) |

AZD1222, Oxford/AstraZeneca vaccine; BNT162b2, Pfizer-BioNTech vaccine; BAME, Black Asian and minority ethnic group; BMI, body mass index; Categorical variables are presented as numbers (percentages), continuous variables were presented as mean± standard deviation or median (inter-quartile range); CMV, cytomegalovirus; CNI, calcineurin inhibitor; CVD, cardiovascular disease; eGFR, estimated glomerular filtration rate; EBV, Epstein bar virus; IS, immunosuppression; MPA, mycophenolic acid; MDRD, modification of diet in Renal disease;.α, excluding non-melanoma skin cancer

The data demonstrate a suboptimal antibody response to SARS-COV-2 vaccination after two vaccine doses in a substantial proportion of KTR. The humoral response rate to the SARS-COV-2 vaccines in our cohort was low at <60% compared to a rate of over 90% in the general population and 99% amongst health care workers [12]. Factors associated with decreased humoral response included treatment with MPA, older age and allograft dysfunction. The number of immunosuppression medications, corticosteroid, CNI, history of cardiovascular disease, diabetes and vaccine type did not influence vaccine response.

This result is consistent with those of previous studies. For instance, Rozen-Zvi et al. reported a seropositivity rate of 36% in a cohort of 308 KTR with an average age of 57 years.

**Table 3. Factors associated with antibody response by univariate, multivariate and propensity score analysis.**

| Univariate Analysis | | | | Multivariate logistic regression | | | | Propensity score Analysis | | | |
|---|---|---|---|---|---|---|---|---|---|---|---|
| Variables | OR | 95% CI | P | Variables | OR | 95% CI | P | Variables | aOR | 95% CI | P |
| Age (Per 10-year increase) | 0.66 | 0.60–0.82 | <0.001 | Age (Per year increase) | 0.61 | 0.48–0.78 | <0.001 | Steroids | 0.64 | 0.28–1.50 | 0.31 |
| Female gender | 1.34 | 0.88–2.04 | 0.17 | BMI | 1.06 | 1.00–1.13 | 0.06 | MPA | 0.17 | 0.07–0.41 | <0.001 |
| Ethnicity | 1.16 | 0.87–1.54 | 0.32 | MPA | 0.02 | 0.01–0.11 | <0.001 | Triple IS | 0.96 | 0.34–2.73 | 0.94 |
| BMI (Per 1mg/kg$^2$) | 1.03 | 0.99–1.06 | 0.18 | Steroid | 0.27 | 0.04–1.97 | 0.20 | Vaccine (AZD1222 vs BNT162b2) | 0.93 | 0.41–2.13 | 0.87 |
| Diabetes | 0.82 | 0.53–1.27 | 0.39 | CNI | 0.20 | 0.02–2.39 | 0.21 | | | | |
| CVD | 0.58 | 0.36–0.93 | 0.03 | Time since transplantation/y | 1.05 | 0.98–1.12 | 0.19 | | | | |
| Primary renal disease | 1.07 | 0.96–1.18 | 0.22 | BK Viremia | 2.58 | 0.83–8.02 | 0.10 | | | | |
| Time since transplantation/y | 1.09 | 1.06–1.14 | <0.001 | eGFR $^{\alpha,\beta}$ | 1.40 | 1.19–1.66 | <0.001 | | | | |
| Cancer | 0.75 | 0.33–1.70 | 0.49 | Vaccine (AZD1222 vs BNT162b2) | 0.58 | 0.30–1.13 | 0.11 | | | | |
| CNI | 0.35 | 0.13–0.95 | 0.04 | Number of IS | 1.64 | 0.25–10.73 | 0.60 | | | | |
| MPA | 0.10 | 0.06–0.20 | <0.001 | | | | | | | | |
| Steroid | 1.53 | 0.99–2.36 | 0.06 | | | | | | | | |
| Number of IS | 0.58 | 0.39–0.86 | 0.007 | | | | | | | | |
| eGFR $^{\alpha,\beta}$ | 1.12 | 1.01–1.25 | 0.03 | | | | | | | | |
| eGFR $\geq 30$ $^\alpha$ | 1.10 | 0.64–1.91 | 0.73 | | | | | | | | |
| eGFR $\geq 60$ $^\alpha$ | 1.90 | 1.16–3.09 | 0.01 | | | | | | | | |
| Donor type | 0.99 | 0.64–1.51 | 0.95 | | | | | | | | |
| BK virus | 2.50 | 1.17–5.33 | 0.02 | | | | | | | | |
| CMV | 0.68 | 0.35–1.31 | 0.25 | | | | | | | | |
| EBV | 0.67 | 0.30–1.50 | 0.33 | | | | | | | | |
| Acute rejection | 2.37 | 1.28–4.38 | 0.006 | | | | | | | | |
| Vaccine (AZD1222 vs BNT162b2) | 0.76 | 0.46–1.27 | 0.29 | | | | | | | | |

$^\alpha$, the Modification of Diet in Renal Disease eGFR;

$^\beta$, per 10 ml/min/1.73m$^2$ increase; AZD1222, Oxford/AstraZeneca vaccine; aOR, adjusted odds ratio; BMI, body mass index; BNT162b2, Pfizer-BioNTech vaccine; CI, confidence interval; CMV, cytomegalovirus; CNI, calcineurin inhibitor; CVD, cardiovascular disease; eGFR, estimated glomerular filtration rate; EBV, Epstein bar virus; IS, immunosuppression; MPA, mycophenolic acid; MDRD, modification of diet in renal disease, OR, odds ratio; y, year

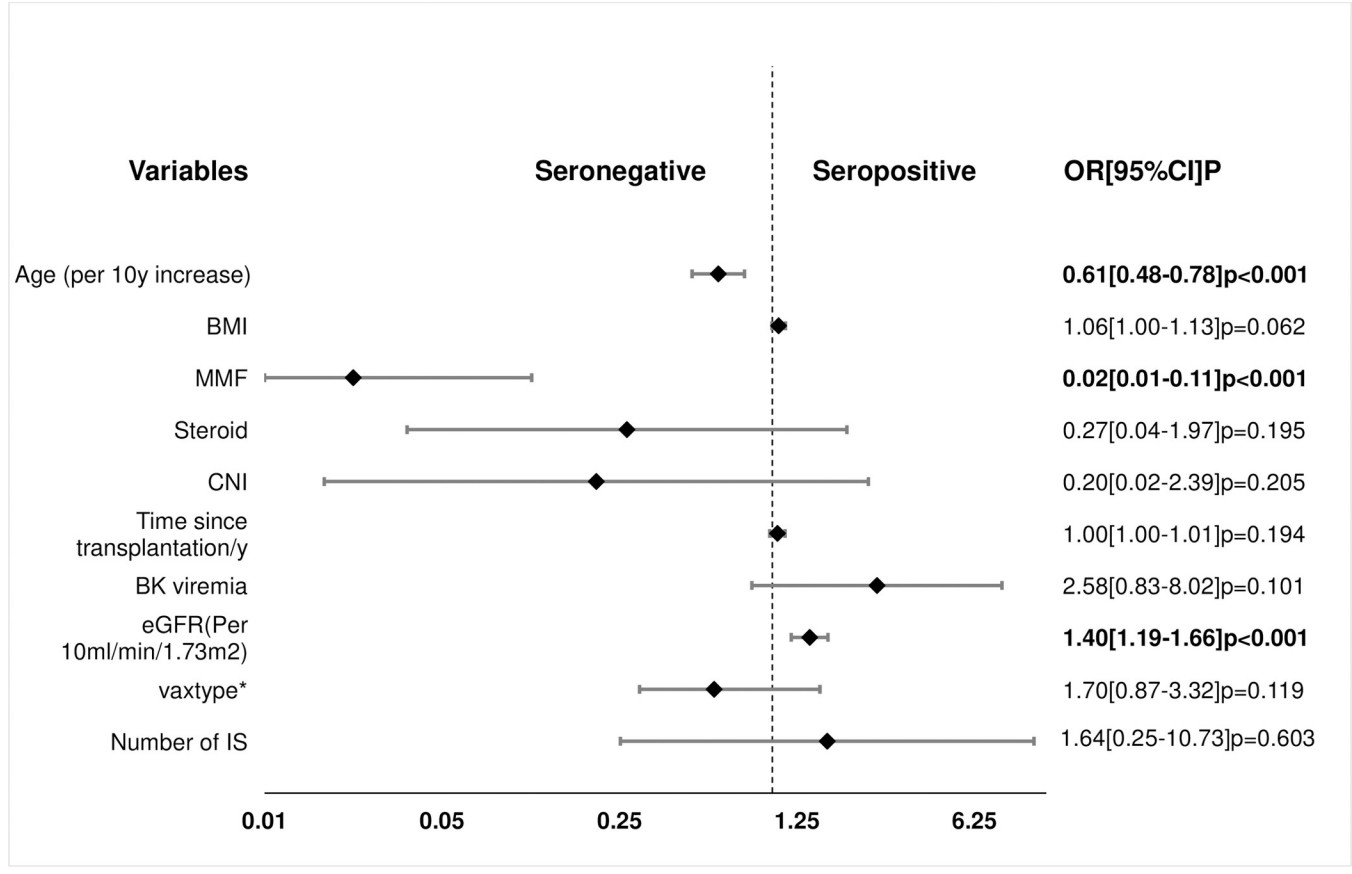

**Fig 2. Factors associated with vaccine response by multivariate analysis.** a, MDRD eGFR mL/min/1.73 m2; CVD, cardiovascular disease; *, AZD1222, Oxford/AstraZeneca vaccine vs BNT162b2, Pfizer-BioNTech vaccine.

The use of MPA and increasing age were significantly associated with decreased odds of antibody response [5]. Boyarsky et al. evaluated 322 KTR who received two doses of SARS-COV-2 reported a seropositivity rate of 48% [13]. In another review of 28 KTR with a median age of 66 years by Husain et al., the anti-SARS-COV-2 seropositivity rate after two vaccine doses was 25% and MPA was significantly associated with poor antibody response [14]. Grupper et al. reported a 37.5% positive response rate among 136 KTR after two doses of the BNT162b2 vaccine. In that study, older age, high-dose corticosteroids, triple immunosuppression and MPA use predicted poor vaccine response [6]. In a recent systematic review by Carr et al., the factors observed to be associated with lack of antibody response among transplant recipients included increasing age, less time since transplant, maintenance with antimetabolites, use of belatacept and triple immunosuppression [3].

Notably, there was a slightly higher antibody response rate in our cohort than previous studies reported (56% compared to 25–48%). One possible reason may be the time interval between transplantation and vaccination. The first 3–6 months post-transplantation is widely known as the period of maximum immunosuppression with evidence pointing to a lower vaccine response during this period of intense immunosuppression [15–17]. A similar response to SARS-CoV-2 vaccines have been reported by recent studies [3, 6, 18–20]. For example, the report of the systematic review by Carr et al, found that less time from transplantation was a strong risk predictor of poor antibody response [3]. The median interval from transplantation

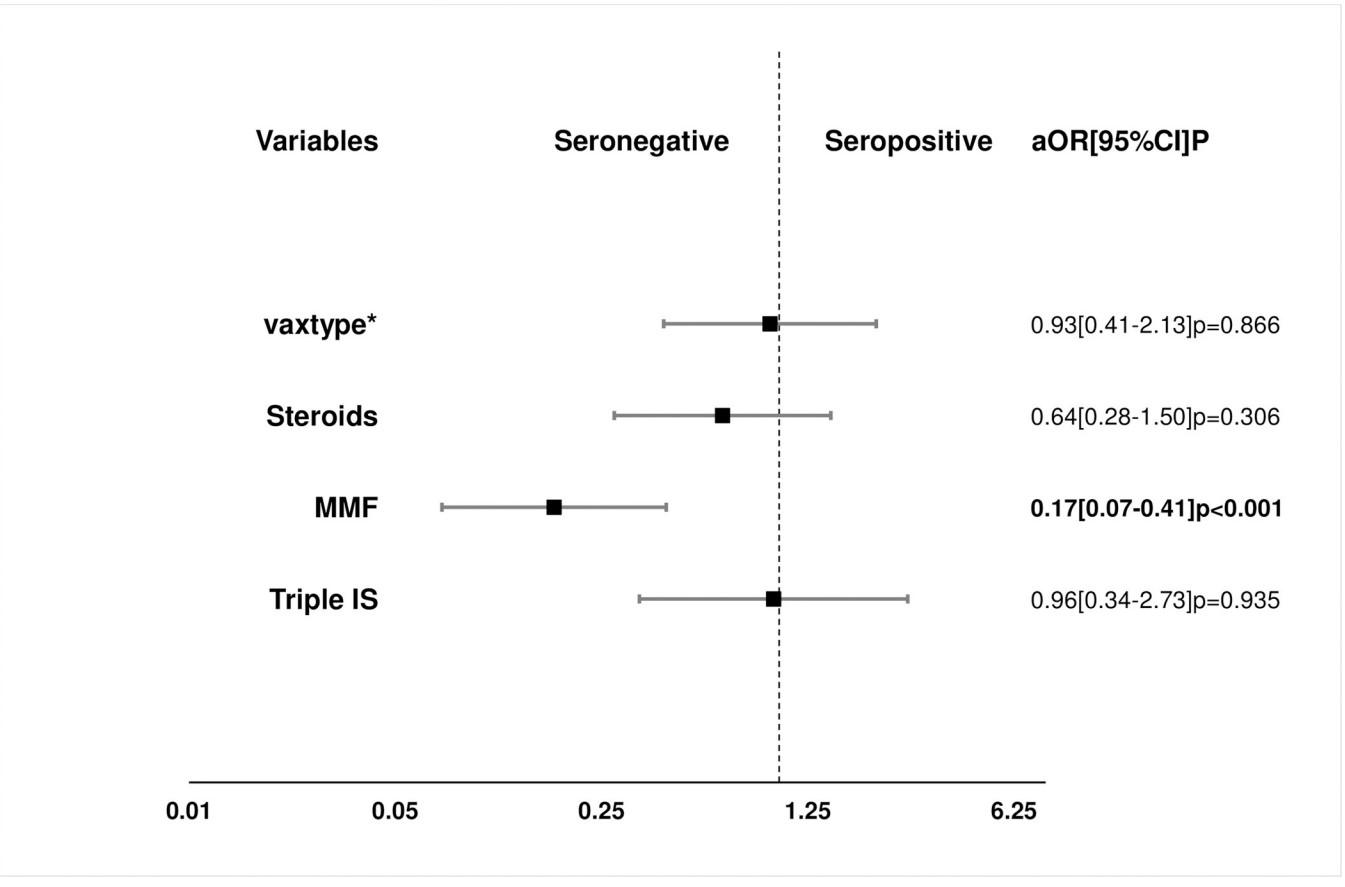

**Fig 3. Propensity scores analysis of the effect of 4 treatment variables on vaccine response.** Propensity score matching was a 1:1 nearest neighbor matching within a propensity score caliper of 0.2 standard deviation of the mean of the logit propensity score; a, MDRD eGFR mL/min/1.73 m$^2$; CVD, cardiovascular disease; $^*$, AZD1222, Oxford/AstraZeneca vaccine vs BNT162b2, Pfizer-BioNTech vaccine; aOR, adjusted odd ratio.

to vaccination in our cohort was 91(48–156) months, suggesting that majority of our cohort was no longer heavily immunosuppressed at the time of vaccination.

Another explanation for this might be the interval between the two vaccine doses. The median interval between the first and second vaccine dose was longer in our cohort (77days) than in other reports (21-31days) [3, 6, 21]. Evidence suggests that vaccine schedules with longer intervals between vaccine doses may improve vaccine responses [22].

As highlighted above, the risk factors for poor antibody response in our cohort were not different from those reported by other studies. These included MPA antimetabolite, increasing age, and lower eGFR.

There is a large body of evidence linking the use of MPA immunosuppression to suboptimal antibody response to vaccines [14, 18, 23–25]. MPA is a potent inhibitor of B-cell function inhibiting the proliferation and differentiation of B-cells by blocking early activation events. This stops the expansion of both naïve and memory B-cells and prevents plasma cell differentiation and antibody production. In addition, MPA also suppresses immunoglobulin secretion from already activated B-cells [26]. In our cohort, we noted over 90% reduction in the odds of seropositivity by MVA and over 80% reduction in the adjusted odd ratio of seropositivity by PSA in patients treated with MPA.

Corticosteroids have also been linked to inadequate antibody response to SARS-COV-2 vaccine [5, 6, 12, 27]. However, this study did not find a significant impact of corticosteroids

on vaccine response. This might be because most KTR in our cohort had their transplant more than 12months before vaccination. As such, their corticosteroid dose would have been tapered down to a minimum maintenance dose of 5mg daily (in the absence of acute rejection episodes) according to our protocol.

It was interesting to note that a previous history of BK viremia was a confounder of antibody response to vaccination. Showing a significant association with a positive antibody response in the univariate analysis but became insignificant when immunosuppression regime was included in the multivariate analysis. This reflects the effect of our protocol for managing BK viremia which involves the reduction or withdrawal of some maintenance immunosuppressive medications most notably the antimetabolite [28, 29].

Another important finding from this study is the link between kidney function and vaccine response. Every 10ml/min/1.73m$^2$ increase in eGFR was associated with a 40% increase in the odds of a positive antibody response. This finding broadly supports reports from other researchers [5, 30]. Mulley et al. found that the likelihood of seroprotection from influenza vaccine was significantly reduced by lower eGFR (OR 0.16) [23]. The findings of Cucchiari et al. also suggest that decreasing eGFR was associated with impaired cellular response to the SARS-COV-2 vaccine in a cohort of 148 kidney transplant recipients [30]. Besides, low kidney function has been shown to impair both the innate and the adaptive immune response with decreased B and T lymphocyte counts, poor lymphocyte activation, impaired monocyte function, inadequate antigen presentation, weakened memory cell generation and low antibody production. These changes become more profound the further the kidney function declines towards CKD 4 and 5 [31]. In addition, endothelial dysfunction, uremic toxins, oxidative stress, mineral bone disease and persistent low-grade inflammation, which accompany renal impairment, exert a profound detrimental effect on vaccine response [31].

Advancing age is widely acknowledged to have a negative influence on vaccine response. It has been shown that influenza and hepatitis B vaccines induce an adequate antibody response in less than half of those >65years [32]. This is thought to be due to a reduced population of naïve B cells, poor antibody responses to protein and polysaccharide antigens and decreased IgG antibody lifespan and diversity with age [32].

One crucial question we set out to answer in this study was whether one vaccine type was superior to the other with respect to antibody response in KTR. This is one of the first reports in kidney transplant recipients to include the AstraZeneca vaccine. However, we found no difference in antibody response to AZD1222, a vector-based SARS-COV-2 vaccine compared to BNT162b2, an mRNA vaccine. Lesny et al., similarly, found no relationship between the vaccine type and antibody response amongst hemodialysis patients receiving either the mRNA or the vector-based SARS-COV-2 vaccines [33]. On the other hand, a few recent studies within and outside the UK have reported higher seroconversion rates and neutralizing antibody titres in recipients of BNT162b2 compared to AZD1222 [34–36]. Of note is a recent sub-study of the UK OCTAVE study in which 920 KTR were screened for spike protein antibodies following 2 doses of either BNT162b2 (n = 490) or AZD1222 (n = 430) vaccines. The result showed a higher seroconversion rate in recipients of BNT162b2 (65.6%) compared to recipients of AZD1222 (43.5%) [34]. Another study that assessed neutralizing antibody levels following COVID-19 vaccination in hemodialysis patients in the UK also reported a suboptimal response to AZD1222. Two studies from Korea respectively evaluated the antibody responses against SARS-CoV-2 in healthcare workers, reported better antibody responses with BNT162b2 compared to AZD1222 [35, 36].

The observed post vaccination COVID-19 infection rates in KTR confirm the clinical importance of the findings of this study. 77% of infections and 100% of deaths post-double vaccination were observed in the seronegative group. There have been multiple reports of

COVID-19 infections after two SARS-COV-2 vaccines [3, 5, 6]. Grupper et al. reported two COVID-19 infections out of 136 fully vaccinated KTR [6], Rosen et al. noted four infections out of 308 vaccinated KTR [5], and Wadei et al. reported five cases of post vaccination COVID-19 infections out of 629 solid organ transplant recipients [7].

One of the strengths of this study is the use of PSA to control for confounding factors while assessing the impact of various treatment factors on vaccine response, minimizing the effects of the lack of randomization on our study. Nevertheless, some limitations of this study need to be acknowledged. Firstly, some of our participants had no pre-vaccination SARS-COV-2 antibody test. Thus, that we cannot confidently exclude previous asymptomatic SARS-COV-2 infections in these participants. However, all our KTR have been closely monitored for COVID-19 symptoms with PCR tests pre and post vaccination. Secondly, the antibody titer was not quantified in this study. Several reports have shown a correlation between the antibody titer and immunity following SARS-COV-2 vaccination [5, 37, 38]. Thirdly, due to variation in follow-up appointment schedules, there was significant variation in the interval between vaccination and antibody test ranging from 8 to 155 days (median = 38days). Fourthly, we did not measure neutralizing antibody activity in our cohort and, as such, cannot make the assertion that a positive antibody test translates into an effective protection against COVID-19 infection in our cohort [3, 39, 40]. Lastly, although eight days was chosen as the minimum interval from vaccination to antibody assay based on the evidence that antibodies develop 1 to 3 weeks post vaccination [41], it is possible that immunocompromised patients such as the subjects in our cohort might take longer than eight days to develop antibodies. However, the median (IQR) time to assay in our cohort was 38(22–55) days. Only 14 subjects had antibody assay less than 14days post vaccination and 6 of these were seropositive.

## Conclusion

Our study has shown a much lower seropositivity rate amongst the KTR after two doses of SARS-COV-2 vaccine than in the general population. Increasing age, use of MPA and lower glomerular filtration rate were factors associated with a non-response. These findings complement those of earlier studies and highlight the need for a tailored approach to the vaccination, post-vaccination surveillance of KTR and make a case for further vaccine doses in this vulnerable group as well as the use of monoclonal antibodies to improve outcomes [42–47].

## Supporting information

**S1 Data.**
(XLSX)

## Author Contributions

**Conceptualization:** Phillip A. Kalra, Dimitrios Poulikakos, Rachel Middleton.

**Data curation:** Chukwuma A. Chukwu, Kassir Mahmood, Safa Elmakki, Julie Gorton.

**Formal analysis:** Chukwuma A. Chukwu.

**Methodology:** Chukwuma A. Chukwu.

**Project administration:** Phillip A. Kalra, Dimitrios Poulikakos, Rachel Middleton.

**Supervision:** Phillip A. Kalra, Dimitrios Poulikakos, Rachel Middleton.

**Validation:** Chukwuma A. Chukwu.

**Writing – original draft:** Chukwuma A. Chukwu.

**Writing – review & editing:** Chukwuma A. Chukwu, Phillip A. Kalra, Dimitrios Poulikakos, Rachel Middleton.

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
