## [Decision Letter · Decision Letter 0]

10 Dec 2021

PONE-D-21-34133Evaluating the antibody response to SARS-COV-2 vaccination amongst kidney transplant recipients at a single nephrology centrePLOS ONE

Dear Dr. Chukwu,

Thank you for submitting your manuscript to PLOS ONE. After careful consideration, we feel that it has merit but does not fully meet PLOS ONE’s publication criteria as it currently stands. Therefore, we invite you to submit a revised version of the manuscript that addresses the points raised during the review process.

ACADEMIC EDITOR: Whilst the data presented in this paper are not particularly novel, they would add to the existing literature on this very timely topic. Authors should clarify a few some of the data presented, and perhaps comment on how this work adds novelty to the literature. I think that the work ought to be published with revisions (if satisfactory) in line with PlosOne policy/philosophy, as the data are sound and the paper is well written.

We look forward to receiving your revised manuscript.

Kind regards,

Frank JMF Dor, M.D., Ph.D., FEBS, FRCS

Academic Editor

PLOS ONE

Journal Requirements:

Reviewers' comments:

Reviewer's Responses to Questions

**Comments to the Author**

1. Is the manuscript technically sound, and do the data support the conclusions?

Reviewer #1: Yes

Reviewer #2: Yes

2. Has the statistical analysis been performed appropriately and rigorously? 

Reviewer #1: Yes

Reviewer #2: Yes

3. Have the authors made all data underlying the findings in their manuscript fully available?

Reviewer #1: Yes

Reviewer #2: Yes

4. Is the manuscript presented in an intelligible fashion and written in standard English?

Reviewer #1: Yes

Reviewer #2: Yes

5. Review Comments to the Author

Reviewer #1: Congratulations on writing this paper. This paper demonstrates poor seroconversion following two doses of either a Pfizer or AstraZeneca SARS-COV-2 vaccine in kidney transplant recipients and the effect mycophenolate, older age and eGFR. Although these are significant findings very similar findings have been reported previously in large cohorts from America, Israel, France and the UK. This is one of the first reports to include the AstraZeneca vaccine although its use was not found to be predictive of seroconversion or seronegativity. The authors claims are properly placed in the context of previous literature. The data and analyses support the claims. There are a few bits of data that I believe need to be clarified. The methods suggest that when the exact date of vaccination was unavailable the date was estimated from the date from the first vaccination. The methods also state to patients had to have a their antibody test at least 8 days from the date of second vaccination. I think it’s important to know how many patients have an estimated date versus an actual date as that may have a significant impact on the results especially given the changes in JCVI recommended timing. The serological testing needs to specify what the antibody was against as we know this has an impact in transplant patients. In this case, the assay used was against the S1 RBD, which appears to perform best in transplant patients. This should be included. The analyses consisting of a univariate analysis, a multivariate analysis and a propensity score analysis appear appropriate. The manuscript is well organised and written clearly.

see attachment for further points

Reviewer #2: Chukwu et al presented a well-written paper on antibody response in kidney transplant recipients to the COVID-19 vaccine. Their retrospective analysis was clearly conveyed and they were transparent in their analysis. While the authors make a great point about the lack of clinical trials studying the COVID-19 vaccine in transplant recipients, their findings are not novel and only contribute to the multitude of papers reporting on the same thing. While there is value in adding to the body of literature on this topic, there should be a more novel angle to their findings as their PSA also did not find anything different. More pertinent studies, such as third vaccine dosing or HLA changes due to antibody responses or revaccination antibody response following transplantation, may have more clinical weight for the readers.

6. PLOS authors have the option to publish the peer review history of their article (what does this mean?). If published, this will include your full peer review and any attached files.

Reviewer #1: No

Reviewer #2: No

---

## [Author Response · Author response to Decision Letter 0]

26 Jan 2022

Please find response attached as a file marked " Response to reviewers"

---

## [Decision Letter · Decision Letter 1]

24 Feb 2022

Evaluating the antibody response to SARS-COV-2 vaccination amongst kidney transplant recipients at a single nephrology centre

PONE-D-21-34133R1

Dear Dr. Chukwu,

We’re pleased to inform you that your manuscript has been judged scientifically suitable for publication and will be formally accepted for publication once it meets all outstanding technical requirements.

Kind regards,

Frank JMF Dor, M.D., Ph.D., FEBS, FRCS

Academic Editor

PLOS ONE

Additional Editor Comments (optional):

Reviewers' comments:

Reviewer's Responses to Questions

**Comments to the Author**

1. If the authors have adequately addressed your comments raised in a previous round of review and you feel that this manuscript is now acceptable for publication, you may indicate that here to bypass the “Comments to the Author” section, enter your conflict of interest statement in the “Confidential to Editor” section, and submit your "Accept" recommendation.

Reviewer #1: All comments have been addressed

Reviewer #2: All comments have been addressed

2. Is the manuscript technically sound, and do the data support the conclusions?

Reviewer #1: Yes

Reviewer #2: Yes

3. Has the statistical analysis been performed appropriately and rigorously? 

Reviewer #1: Yes

Reviewer #2: Yes

4. Have the authors made all data underlying the findings in their manuscript fully available?

Reviewer #1: Yes

Reviewer #2: Yes

5. Is the manuscript presented in an intelligible fashion and written in standard English?

Reviewer #1: Yes

Reviewer #2: Yes

6. Review Comments to the Author

Reviewer #1: my comments have been incorporated-

................................................................

Reviewer #2: The conclusion is much more comprehensive and has hit upon the issues brought up in the prior review.

7. PLOS authors have the option to publish the peer review history of their article (what does this mean?). If published, this will include your full peer review and any attached files.

Reviewer #1: No

Reviewer #2: No

---

## [Editor Report · Acceptance letter]

1 Mar 2022

PONE-D-21-34133R1 

Evaluating the antibody response to SARS-COV-2 vaccination amongst kidney transplant recipients at a single nephrology centre 

Dear Dr. Chukwu:

I'm pleased to inform you that your manuscript has been deemed suitable for publication in PLOS ONE. Congratulations! Your manuscript is now with our production department. 

Kind regards, 

on behalf of

Dr. Frank JMF Dor 

Academic Editor

PLOS ONE